# Clinician Assessment Tools for Patients with Diabetic Foot Disease: A Systematic Review

**DOI:** 10.3390/jcm9051487

**Published:** 2020-05-15

**Authors:** Raúl Fernández-Torres, María Ruiz-Muñoz, Alberto J. Pérez-Panero, Jerónimo C. García-Romero, Manuel Gónzalez-Sánchez

**Affiliations:** 1Department of Nursing and Podiatry, University of Málaga, Arquitecto Francisco Peñalosa, s/n. Ampliación Campus de Teatinos, 29071 Málaga, Spain; raulft.95@gmail.com (R.F.-T.); albertoj2p@hotmail.com (A.J.P.-P.); 2Medical School of Physical Education and Sports, University of Málaga, C/Jiménez Fraud 10. Edificio López de Peñalver, 29010 Málaga, Spain; jeronimo@uma.es; 3Department of Physiotherapy, University of Málaga, Arquitecto Francisco Peñalosa, s/n. Ampliación campus de Teatinos, 29071 Málaga, Spain; mgsa23@uma.es

**Keywords:** diabetes, diabetes complications, ulcer, outcome measures, evidence, review

## Abstract

The amputation rate in patients with diabetes is 15 to 40 times higher than in patients without diabetes. To avoid major complications, the identification of high-risk in patients with diabetes through early assessment highlights as a crucial action. Clinician assessment tools are scales in which clinical examiners are specifically trained to make a correct judgment based on patient outcomes that helps to identify at-risk patients and monitor the intervention. The aim of this study is to carry out a systematic review of valid and reliable Clinician assessment tools for measuring diabetic foot disease-related variables and analysing their psychometric properties. The databases used were PubMed, Scopus, SciELO, CINAHL, Cochrane, PEDro, and EMBASE. The search terms used were foot, ankle, diabetes, diabetic foot, assessment, tools, instruments, score, scale, validity, and reliability. The results showed 29 validated studies with 39 Clinician assessment tools and six variables. There is limited evidence on all of the psychometric characteristics of the Clinician assessment tools included in this review, although some instruments have been shown to be valid and reliable for the assessment of diabetic neuropathy (Utah Early Neuropathy Scale or UENS); ulceration risk (Queensland High Risk Foot Form or QHRFF); diabetic foot ulcer assessment, scoring, and amputation risk (Perfusion, extent, depth, infection and sensation scale or PEDIS and Site, Ischemia, Neuropathy, Bacterial Infection, and Depth score or SINBAD); and diabetic foot ulcer measurement (Leg Ulcer Measurement Tool LUMT).

## 1. Introduction

Diabetes is a chronic disease that might lead to several systemic complications, such as diabetic foot disease (DFD), a common condition with a global prevalence of 6.3%, which is known to affect wound onset and healing [1]. Once a diabetic foot ulcer (DFU) occurs, treatment is often unsuccessful due to infection or neurovascular complications. Patients often require a minor or major amputation, which negatively affects their quality of life and survival [2,3]. The amputation rate in the diabetic population is 15 to 40 times higher than in non-diabetic patients [4]; therefore, DFD translates into high social impact and poor clinical prognosis, creating a heavy burden for health services [5]. The identification of high-risk diabetic patients through early assessment remains the primary action to avoid major complications [6]. For this purpose, valid and reliable instruments are required in order to assess diabetic foot-related variables.

Outcome evaluations measure how patients feel, function, or survive after a disease or condition. The evaluations include clinical outcome assessments (COAs) and biomarkers. Biomarkers are based on an organic process, whereas COAs are based on the implementation and/or interpretation of information from a patient. There are four types of COAs: patient-reported outcomes (PROs), clinician assessment tools or scales, observer-reported outcomes, and performance outcomes [7].

For clinician assessment tools, clinical examiners must be specifically trained to make a correct judgment that is based on patient outcomes. This assessment helps to efficiently identify at-risk patients and monitor whether or not they should undergo an intervention [7]. To our knowledge, there are no scientific reports that thoroughly review DFD clinician assessment tools and their psychometric properties. Previous studies have investigated DFU assessment scales [8,9], but none have performed a detailed analysis of psychometric properties or included scales that measure factors other than the presence and characteristics of an ulcer (diabetic neuropathy, ulcer risk).

Therefore, the aims of this work were, as follows: (1) to identify valid and reliable clinician assessment tools for measuring DFD-related variables, (2) to analyse the psychometric properties of the identified clinician assessment tools or scales, and (3) to highlight clinician assessment tools that are associated with improved recommendations for patients with DFD based on their psychometric properties.

## 2. Materials and Methods

### 2.1. Protocol and Registration

This systematic review was developed following the preferred reporting items for systematic reviews and meta-analyses (PRISMA) guidelines [10]. It was registered in the PROSPERO database (CRD no. 42019118202).

### 2.2. Eligibility Criteria

The study subjects were patients with DFD, which included those with diabetic foot (infection, ulceration, or destruction of tissues of the foot of a person that is diagnosed with diabetes mellitus) or diabetic neuropathy (the presence of symptoms or signs of nerve dysfunction in a person with diabetes mellitus) [11], regardless of the type of diabetes, intervention, gender, or age. All the studies of clinician assessment tools for DFD evaluation and monitoring were included. All documents published up to 30 December 2019 that were published in English, French, Italian, Portuguese, or Spanish were included. Studies of clinician assessment tools that did not include any psychometric properties in their development or did not provide any measurement properties that met the consensus-based standards for the selection of health measurement instruments (COSMIN) criteria were excluded [12]. Although the COSMIN methodology is aimed at studies on the measurement properties of PROMs, it can also be used in other types of studies on the measurement properties. These measurement properties are reliability (internal consistency, reliability, and measurement error), validity (content validity, construct validity, and criterion validity), and responsiveness [12].

### 2.3. Sources and Search

The databases used were PubMed, Scopus, SciELO, CINAHL, Cochrane, PEDro, and EMBASE. The following search words were used: foot, ankle, diabet *, diabetic foot, assessment, tools, instruments, score, scale, valid *, and reliab *.

Taking into account the differences between each database, the following search strategy was used: (((foot or ankle) AND (diabet *)) OR (diabetic foot)) AND ((assess *) OR (scale) OR (score) OR (instruments) OR (valid *) OR (reliab *)).

### 2.4. Study Selection

Three reviewers independently reviewed all of the articles and participated in each phase of study selection. Before reading the full-text articles, the titles and abstracts of the references that were identified in the initial search were filtered according to the eligibility criteria. Differences in judgment were resolved by agreement between the reviewers.

### 2.5. Data Extraction and Synthesis

In order to facilitate understanding of the results, variables measured by clinician assessment tools were classified into two categories according to diagnostic purpose: “DFD-related variables” and “DFU-related variables”.

The COSMIN checklist is considered to be useful for reporting study measurement properties; therefore, the COSMIN measurement property taxonomy was followed to synthesise the results of this review [12]. Nevertheless, the data extraction process was adapted to take into account all psychometric properties, as this study aimed to review all clinician assessment tools. The following measurements were included in the synthesis of results: sensitivity, specificity, positive predictive value (VPP), negative predictive value (VPN), positive probability ratio (LR+), negative probability ratio (LR−), area under the receiver operating characteristic curve (AUC-ROC), gold standard, agreement with gold standard, inter-rater reliability(INTER), and intrarater reliability (INTRA). Other data were extracted in order to complete the general analysis of each study: taxonomy of the clinician assessment tool, authors, publication year, type of diabetes, and number of study subjects.

## 3. Results

The flow diagram (Figure 1) summarises the study selection stages for the studies that were included in this review, detailing the reasons for exclusions [10]. The most common reasons for exclusion of full-text articles were publication in a language different from those that were included in the eligibility criteria or not reporting any psychometric or measurement properties.

In total, 29 validated studies with 39 scales and six variables were included. The group “DFD-related variables” contained two variables (diabetic neuropathy, ulcer risk) and the group “DFU-related variables” contained four variables (amputation risk, DFU healing, DFU infection assessment, and DFU measurement).

The validation study that included the largest sample was the one that validated the Curative Health Services system (CHS) (*n* = 19,280) and the one with the smallest sample validated the Leg Ulcer Measurement tool (LUMT) (*n* = 22).

The most popular validation method for clinician assessment tools or scales was the calculation of psychometric properties that were based on 2 × 2 contingency tables, used in 17 out of 29 studies. Four studies validated clinician assessment tools following the indications in the COSMIN guidelines, while eight studies used different statistical calculations. Sensitivity and specificity were the most frequently calculated psychometric properties, with their value being known for 29 and 27 scales, respectively. PPV was calculated for 21 scales, NPV for 19 scales, LR+ for 14 scales, and LR− for 13 scales. Inter- and intrarater reliability were the least often calculated (10 and seven scales, respectively).

Table 1 shows the DFD-related variables and the validated clinician assessment tools for measurement. Two variables and 20 scales were included in this category. Ten scales were found for the assessment of diabetic neuropathy and 10 for the assessment of ulceration risk.

Table 2 shows the DFU-related variables and the validated clinician assessment tools. Three variables and 17 scales were included in this category. Most of the scales (*n* = 10) were related to DFU scoring, assessment, and risk of amputation.

Table 3 shows the DFD- and DFU-related variables that followed the COSMIN guidelines [9] in order to measure the validity and reliability. These five scales, unlike those that are presented in Table 1 and Table 2, do not calculate properties, like sensitivity or specificity.

An additional table was included (Appendix A) with other details of each scale, including the study subjects, dimensions, punctuation, and description.

## 4. Discussion

This work aimed to identify validated clinician assessment tools for the measurement of DFD-and DFU-related variables, in order to analyse their psychometric properties and highlight those scales with the best examples. For most outcome variables, at least one scale has been highlighted for its psychometric properties, which have been expanded on in two additional sections at the end of the discussion, namely recommendations for clinical application and potential developments from a research perspective. The objectives were accomplished based on the results that were obtained in this study.

In the present study, the psychometric characteristics of all clinician assessment and follow-up tools in patients with DFD or DFU are presented. However, when interpreting the results, it is important to consider that there are a series of conditioning factors that limit the ability make comparisons between different tools or scales. We observed that some psychometric properties present better results in one scale than another [42,43,44].

The difference in results could be justified in multiple ways, all of which fall into two basic categories. On the one hand, there are the structural characteristics of the scale itself, aspects, such as the number of items that make up the tool and the way in which each of those items is evaluated, which are two aspects that definitively determine the way in which the clinician assessment tool is used [44,45]. On the other hand, the conditions in which the scale is used unequivocally determine its possible result. In this sense, it is also important to take the profile of the person being evaluated into account. Thus, descriptive characteristics, such as age, gender, time at which the pathology began, education level, socioeconomic aspects and sociocultural aspects, among others, how the final result of the evaluation might determine whether a specific variable is seen [42,45,46].

In this analysis, we aimed to carry out an exhaustive comparison. However, comparisons could only be performed when the conditions for the assessment process and the subject profile were the same, as the structure of the tools cannot be changed; that is, the conditions and descriptive characteristics of the sample were similar. Only under these conditions could we perform a comparison that minimises the error that is derived from the divergence in conditions or profiles of the subjects evaluated [43,44,47,48]. This is important to keep in mind when selecting a clinician assessment tool, as health professionals or researchers should select a tool that used similar conditions and subject profiles for its validation to the sample being assessed [45,46,47].

The number of subjects that were used to validate the tool is a final aspect that must be taken into account. Some scales have very good psychometric characteristics, but the sample used for validation was too small. In this case, the results should be interpreted as a possible tendency to produce results, as they do not have the necessary strength to be considered 100% valid and reliable. Type 2 (beta) errors, resulting from an inadequate sample size to achieve results with the necessary robustness, is an area frequent limitation in many validation studies of clinician assessment tools [44,45].

### 4.1. Recommendations for Research

The most common validation method in the studies reviewed (17 out of 29) was the use of gold standards togenerate 2 × 2 contingency tables. This suggests a tendency to submit clinician assessment tools or scales to external validation that does not follow the COSMIN guidelines, used in four studies in this review. It is recommended that a criterion for scale validation that uses the same methodology is established in order to facilitate dissemination, understanding, and comparison of results.

The calculation of psychometric properties that were derived from sensitivity and specificity was found to be lacking PPV and NPV, which reflect the impact of prevalence on validity [43,49], and +/−LR were important in terms of the probability of the score to detect true positives and negatives [44,50]. Therefore, we recommend that all of the psychometric properties should be calculated in future validation studies.

Likewise, inter- and intrarater reliability were frequently absent in validation studies, which are fundamental properties for determining whether or not the scales should be recommended for that purpose [45,46,51,52]. This is highly dependent on the criteria and skills of the clinician or researcher. Similarly, the responsiveness was only known for one scale [40], leading to missed information regarding the sensitivity to change for the vast majority of scales evaluated in this review.

Only 12 validation studies specified the type of diabetes of subjects in their sample. This might have an impact on aspects, such as diabetic neuropathy [47,53], so it seems necessary to include in future studies.

### 4.2. Clinical Recommendations

When deciding which tool or scale to use in a clinical setting, it is important to make two considerations. On the one hand, it is necessary to take the characteristics of the clinical setting, the type of patients, and the variables that will be evaluated, among other factors, into account [42,45]. On the other hand, clinical staff must choose an accurate and adequate assessment tool so that the quality of the results is guaranteed in order to guarantee the quality of the results. The number of available questionnaires to evaluate similar variables has increased exponentially in recent years [42,45,47]. However, many questionnaires that are used in the clinical setting have not undergone a comprehensive validation process [42,48]. Therefore, it is important to know the scale that is intended to be used as the process of evaluation and monitoring of the patient depends, at least in part, on the selection of the correct tool [43,44].

The information that an assessment and monitoring tool offers also depends to some extent on its psychometric characteristics. Clinical staff with little training in the selection of evaluation tools should ask the following question: what are the most important psychometric characteristics when evaluating an evaluation and monitoring tool? Validity and reliability are the two characteristics that researchers unanimously identify to be the most decisive [42,43,44,47,48]. The validity of a tool refers to its ability to be used to measure the variable for which it was designed, the more precise the better [54]. Several types of validity that can be analysed, but they most commonly include construct validity, content validity, and criterion validity [44]. On the other hand, reliability is defined as the ability of an assessment tool to provide consistent results over time when the assessment conditions are the same, including the administration method, characteristics of the sample, type of instrument and evaluators, among others. From its definition, it can be easily deduced that reliability is a changing characteristic within the valuation instrument, which will depend, to a great extent, on the conditions in which it is used. This is why it is necessary to carry out a new validation study when the evaluation conditions change considerably [43,45,47,54].

These two variables are not completely independent, as an assessment tool that is not reliable cannot be valid, but an assessment tool might be reliable, but not valid [42,44,45,46,54].

Finally, although the reliability and validity of an assessment tool are the two most important psychometric characteristics, it is important not to forget that there are characteristics that are related to responsiveness, especially in the clinical setting, defined as the ability of an assessment tool to identify changes that may occur over time (as a consequence of treatment, for example) [54].

Based on the results of this review, it would be suitable to use the Leg Ulcer Measurement Tool scale (LUMT) for ulcer measurement and theQueensland High Risk Foot Formscale (QHRFF) to assess ulceration risk. However, the psychometric characteristics of these tools do not have sufficient strength, because they did not use a sufficient sample [44], as the LUMT and QHRFF validation studies were carried out with 19 and 22 subjects, respectively, as previously noted. The Perfusion, extent, depth, infection and sensation scale or PEDIS and the Site, Ischemia, Neuropathy, Bacterial Infection, and Depth score (SINBAD) scales for DFU assessment may be recommended over other clinician assessment tools or scales due to their poor intrarater reliability. Similarly, the Utah Early Neuropathy Scale (UENS) for diabetic neuropathy assessment should be used with caution, due to the unavailable data on intrarater reliability. The results of this review are not conclusive for recommending a scale to assess DFU healing due to their variability issues. Even though it was not discussed in the reviewed studies, it seems accurate to consider, or ideally measure, the time that it takes to complete each scale in order to make a definitive recommendation, as this might alter the normal development of a health service’s attention.

It would be advisable to combine scales in patients with diabetic foot with PRO scales and objective measurements with OCOMs and/or biomarkers in order to obtain a multidisciplinary and complete approach to patient monitoring and follow-up. Moreover, using any of these three approaches is not exclusive to others (e.g., a DFU with a low risk of amputation in a patient with high HbAc1 levels, a sedentary lifestyle and bad habits), as they all play an important role in the assessment and monitoring of patients with DFD.

The present study has a number of limitations that must be taken into account when interpreting the results. Firstly, although an attempt was made to carry out the widest possible search from a linguistic point of view, including five languages (English, French, Spanish, Portuguese, and Italian), we might have missed other tools published in a language other than those mentioned. Secondly, in the present study, an enormous effort was made to synthesise the most frequently studied psychometric characteristics; however, there are other psychometric characteristics that were not included in this study.

## 5. Conclusions

Although there is limited evidence for the psychometric characteristics of the clinician assessment tools that are included in this review, some instruments were found to be valid and reliable for the assessment of diabetic neuropathy (UENS), ulceration risk (QHRFF); DFU assessment, scoring, and amputation risk (PEDIS and SINBAD); and, DFU measurement (LUMT). However, further research is required to improve the reliability and validity of properties in the recommended scales from this review, according to the evidence provided in the literature.

## Figures and Tables

**Figure 1 jcm-09-01487-f001:**
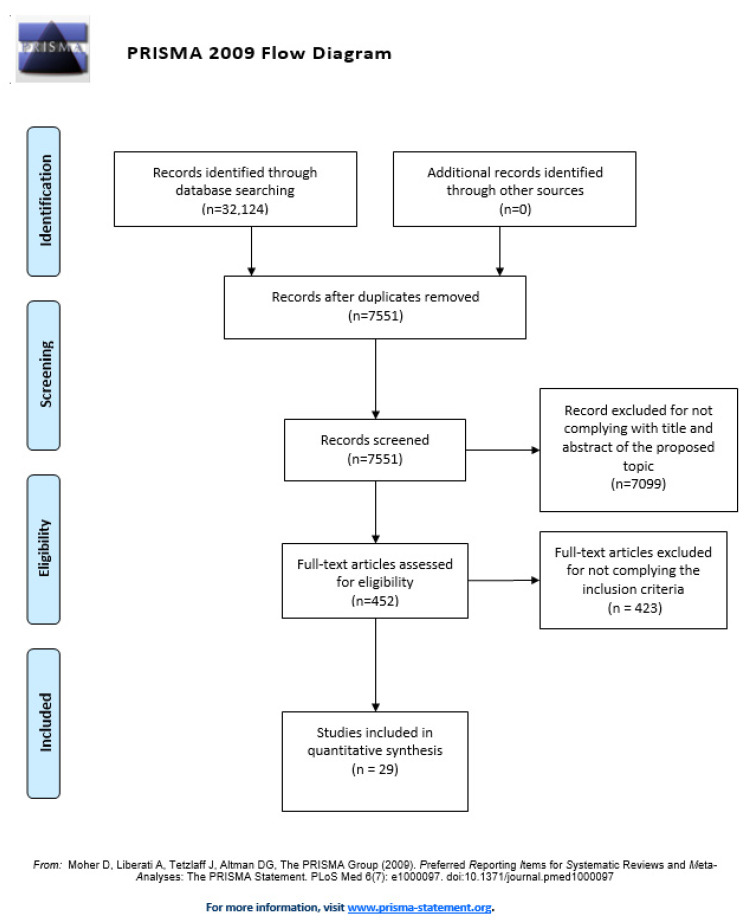
PRISMA Flow Diagram adapted with permission from The PRISMA group, 2020 [10].

**Table 1 jcm-09-01487-t001:** Clinician assessment tools validated for the assessment of diabetic foot disease (DFD) related variables.

Variable	Scale	AUT (Year)	Type	*n*	SENS (%)	SPEC (%)	PPV (%)	NPV (%)	LR+	LR−	AUC-ROC (%)	Gold Standard (GS)	Agreement with GS	Inter	Intra
Diabetic Neuropathy assessment	Early Neuropathy Scale (ENS)	Zilliox et al. (2015) [13]	_	113	83	97	99	67	26.67	0.17	0.94 to 0.96	Nerve Conduction Studies (NCS), Quantitative Sensory Testing, Sudomotor Axon Reflex, Intraepidermal Nerve Fiber Density.	_	_	_
	Total Neuropathy Score (Clinical)	Zilliox et al. (2015) [13]	_	113	81	97	99	66	25,9	0.2	0.97 to 0.99	Same as above	_	_	_
	Modified Toronto Clinical Neuropathy Scale (mTCNS)	Zilliox et al. (2015) [13]	_	113	98	97	99	94	31.2	0.03	1	Same as above	_	_	_
	Neuropathy Impairment Score in the Lower Limbs (NIS-LL)	Zilliox et al. (2015) [13]	_	113	83	97	98	69	26.47	0.18	0.94 to 0.96	Same as above	_	_	_
	Toronto Clinical Neuropathy Scale (TCNS)	Bril et al. (2002) [14]	1/2	89	-	-	-	-	-	-	-	Sural Nerve Fiber Density	*R*^2^ = 0.256		
	Neuropathy disability score (NDS)	Asad et al. (2010) [15]	2	60	92.31	47.62	76.6	76.92	_	_	_	NCS	_	_	_
	Diabetic Neuropathy Examination (DNE)	Asad et al. (2010) [15]	2	60	17.95	100	100	39.62	_	_	_	NCS	Accuracy = 46.67%		
	United Kingdom Screening Test (UKST)	Fateh et al. (2016) [16]	1/2	125	63.93	50	_	_	1.28	0.72	_	NCS	_	_	_
	Michigan Neuropathy Screening Instrument (MNSI)	Fateh et al. (2016) [16]	1/2	125	75.21	33.3	_	_	1.13	0.74	_	NCS	_	_	_
	Utah Early Neuropathy Scale (UENS)	Singleton et al. (2008) [17]	_	215	92	_	_	_	_	_	0.88	NCS	*r* = 0.298 to −0.401	ICC = 0.94	
Ulceration risk	60-s Inlow’s assessment tool	Murphy et al. (2012) [18]	_	69	_	_	_	_	_	_	_	Development of DFU	_	ICC = 0.83 to 0.93	ICC = 0.96 to 1.00
	Basic Foot Screening Checklist (BFSC)	Bower et al. (2009) [19]	2	500	54	77	82	_	_	_	_	Modified Royal Perth Hospital Podiatry Department’s neurovascular assessment tool	_	*k* = 0.35	_
	Queensland High Risk Foot Form (QHRFF)	Lazzarini et al. (2014) [20]	1/2	Intra = 19 Inte*r* = 43 SENS PPV = 32	88 to 100	_	88	_	_	_	_	Development of DFU		*k* = 1.00	*k* = 1.00
	Scottish Foot Ulcer Risk Score (SFURS)	Leese et al. (2006) [21]	_	3526	84.3 to 95.2	66.8 to 90	29.4	99.6	_	_	_	Development of DFU		_	_
	Diabetic foot ulceration risk checklist (DFURC)	Zhou et al. (2018) [22]	1/2	477	62	75	_	_	_	_	0.77	Development of DFU	*r* = 0.76	_	_
	American Diabetes Association System(ADA)	Monteiro-Soares et al. (2012) [23]	_	364	90.9 to 100	13 to 70.4	10.3 to 23.4	98.7 to 100	1.1 to 3.1	0.1	0.83	Development of DFU	_	_	_
	Modified International Working Group on the Diabetic Foot System (IWGDF)	Monteiro-Soares etal. (2012) [23]	_	364	87.9 to 100	38.4 to 70.7	13.9 to 23	98.3 to 100	1.6 to 3	0.2	0.86	Development of DFU	_	_	_
	University of Texas System (UT)	Monteiro-Soares et al. (2012) [23]	_	364	57.6 to 72.7	65.9 to 84.6	17.5 to 27.1	95.2 to 96	2.1 to 3.7	0.4 to 0.5	0.73	Development of DFU	_	_	_
	Scottish Intercollegiate Grouping Network System (SIGN)	Monteiro-Soares et al. (2012) [23]	_	364	100	8.7 to 51.4	9.9 to 17	100	1.1 to 2.1	NC	0.75	Development of DFU	_	_	_
	Seattle Risk Score	Monteiro-Soares et al. (2012) [23]	_	364	69.7 to 93.9	43.8 to 83.4	14.3 to 29.5	96.5 to 97.9	1.7 to 4.2	0.1 to 0.4	0.82	Development of DFU	_	_	_

AUT (Year) = authors of the validation study and year of publication; Type = type of diabetes; SENS = sensitivity; SPEC = specificity; PPV = positive predictive value; NPV = negative predictive value; LR+ = positive likelihood ratio; LR− = negative likelihood ratio; AUC-ROC = area under the receiver operator characteristic curve; Gold Standard = gold standard used for external validity; Agreement with GS = degree of external validity with the gold standard; Inter-Rater = inter-rater reliability; Intra-Rater = intra-rater reliability; NC = non-calculable; ICC = Intraclass Correlation Coefficient; *k* = Cohen’s kappa coefficient.

**Table 2 jcm-09-01487-t002:** Clinician assessment tools validated for the assessment of diabetic foot ulcer (DFU) related variables.

Variable	Scale	AUT (Year)	Type	*n*	SENS (%)	SPEC (%)	PPV (%)	NPV (%)	LR+	LR−	AUC-ROC (%)	Gold Standard (GS)	Agreement With GS	Inter	Intra
DFU assessment scoring and amputation risk	Diabetic Ulcer Severity Score (DUSS)	Monteiro-Soares et al. (2014) [24]	_	137	84	69	72	81	_	_	0.8065	Amputation prediction	Accuracy = 76%	_	_
	Depth, extent of bacterial colonization, phase of healing and associated etiology (DEPA)	Monteiro-Soares et al. (2014) [24]	_	137	79	84	83	81	_	_	0.8908	Amputation prediction	Accuracy = 82%	_	_
	Site, Ischemia, Neuropathy, Bacterial Infection, and Depth score (SINBAD)	Monteiro-Soares et al. (2014) [24]	_	137	63	91	88	72	_	_	0.8483	Amputation prediction	Accuracy = 77%	_	_
		Forsythe et al. (2016) [25]	1/2	37	_	_	_	_	_	_	_	_	_	ICC = 0.91	ICC = 0.44
	Wagner’s classification	Monteiro-Soares et al. (2014) [24]	_	137	75	94	93	80	_	_	0.8921	Amputation prediction	Accuracy = 85%	_	_
		Bravo-Molina et al. (2016) [26]	1/2	250	_	_	_	_	_	_	_	_	_	*k* = 0.55	_
	Perfusion, extent, depth, infection and sensation scale (PEDIS)	Chuan et al. (2015) [27]	1/2	364	93	82	_	_	_	_	0.95	Healing, unhealing and amputation	_	_	_
		Bravo-Molina et al. (2016) [26]	1/2	250	_	_	_	_	_	_	_	_	_	*k* = 0.574	_
		Forsythe et al. (2016) [25]	1/2	37	_	_	_	_	_	_	_	_	_	ICC = 0.80 to 0.90	ICC = 0.23 to 0.42
	University of Texas classification	Armstrong et al. (1998) [28]	_	360	_	_	_	_	_	_	_	Amputation prediction	R^2^= 143.1 and 91	_	_
		Bravo-Molina et al. (2016) [26]	1/2	250	_	_	_	_	_	_	_	_	_	*k* = 0.513	_
		Forsythe et al. (2016) [25]	1/2	37	_	_	_	_	_	_	_	_	_	ICC = 0.94	ICC = 0.53
	Diabetic Foot Ulcer Assessment Scale (DFUAS)	Arisandi et al. (2016) [29]	2	62	89	71	86	77	3.11	0.16	0.9	Total score of BWAT, PUSH and wound surface area	*r* = 0.83 to 0.92	ICC = 0.98	_
	Photographic Wound Assessment Tool (PWAT)	Thompson et al. (2013) [30]	_	68 *	_	_	_	_	_	_	_	Bedside assessments	ICC = 0.89	ICC = 0.40 to 0.80	ICC = −0.24 to 0.95
	Diabetic foot risk assessment (DIAFORA)	Monteiro-Soares et al. (2016) [31]	_	293	57 to 100	53 to 88	11 to 58	88 to 100	2 to 5	0.03 to 0.5	0.91	Amputation prediction	_	_	_
DFU healing	Chronic Lower Extremity Ulcer Score	Beckert et al. (2009) [32]	_	2019	_	_	_	_	_	_	_	Healing time	Kaplan-Meier = 0.45 to 0.83	_	_
	Pressure Ulcer Scale for Healing (PUSH)	Gardner et al. (2011) [33]	_	29	_	_	_	_	_	_	_	Healing time	R^2^ = 0.76	_	
	Curative Health Services system (CHS)	Margolis et al. (2003) [34]	_	19280	_	_	_	_	0.48 to 3.84	_	0.65 to 0.70	Wound closure at week 20	_	_	_
	Sepsis, arteriopathy, denervation System (SAD)	Parisi et al. (2008) [35]	-	105	87.5	52.2	65	80	_	_	_	Healing time	Accuracy = 70.2%	_	_
DFU infection assessment	Non healing, exudates, red tissue, debris, smell criteria (NERD)	Woo et al. (2009) [36]	_	112	32 to 60	47 to 86	_	_	_	_	_	Microbiological analysis	_	_	_
	Size, temperature, osteomyelitis edema, exudate, smell criteria (STONEES)	Woo et al. (2009) [36]	_	112	37 to 87	44 to 89	_	_	_	_	_	Microbiological analysis	_	_	_
	Clinical Signs and Symptom Checklist (CSSC)	Gardner et al. (2009) [37]	1/2	64	0 to 88	0 to 91	_	_	_	_	0.38 to 0.56	Microbial load	_	_	_
	Infectious Diseases Society of America classification (IDSA –IWGDF)	Lavery et al. (2007) [38]	_	1666	_	_	_	_	_	_	_	Amputation and lower extremity-related hospitalization risk	*R*^2^ = 108 to 118.6	_	_

AUT (Year) = authors of the validation study and year of publication; Type = type of diabetes; SENS = sensitivity; SPEC = specificity; PPV = positive predictive value; NPV = negative predictive value; LR+ = positive likelihood ratio; LR− = negative likelihood ratio; AUC-ROC = area under the receiver operator characteristic curve; Gold Standard = gold standard used for external validity; Agreement with GS = degree of external validity with the gold standard; Inter-Rater = inter-rater reliability; Intra-Rater = intra-rater reliability; ICC = Intraclass Correlation Coefficient; *k* = Cohen’s kappa coefficient. * venous/arterial leg wounds (n = 13), diabetic foot wounds (n = 18), pressure ulcers (n = 32), and wounds of other etiologies (n = 5).

**Table 3 jcm-09-01487-t003:** Clinician assessment tools validated for the assessment of DFD and DFU related variables according to consensus-based standards for the selection of health measurement instruments (COSMIN) guidelines.

					Reliability					Validity				Responsiveness
					Internal Consistency	ICC		SEM		Content Validity	Construct Validity	Criterion Validity		
Variable	Scale	Aut (Year)	Type	*n*		Inter	Intra	Inter	Intra			Concurrent	Predictive	
Diabetic neuropathy assessment *	Toronto Clinical Neuropathy Score(TCNS)	Bril et al. (2009) [39]	1/2	65	0.76	0.83	*k* = 0.62 to 1.00	_	_	_	_	_	_	_
	Modified Toronto Clinical Neuropathy Score (mTCNS)				0.78	0.87	*k* = 0.54 to 0.73	_	_	_	*R* = −0.45 to 0.82	_	_	_
DFU measurement **	Leg Ulcer Measurement Tool (LUMT)	Woodbury et al. (2004) [40]	_	22	_	0.77 and 0.89	0.96	3.3 and 4.8	2.0	_	*R* = 0.43 and 0.82	*R* = 0.82	_	0.84
DFU assessment, scoring and amputation risk **	Diabetic foot ulcer assessment scale (DFUAS)	Arisandi et al. (2016) [29]	_	66	_	0.98	_	_	_	Done	*P* < 0.001	*R* = 0.83 to 0.92	SENS = 89% SPEC = 71% PPV = 86% NPV = 77%	_
DFU infection assessment **	Diabetic Foot Infection Wound Score (DFIWS)	Lipsky et al. (2009) [41]	_	371	0.70 to 0.95	_	_	_	_	_	*R* = 0.21 and 0.27	_	_	_

AUT (YEAR) = authors of the validation study and year of publication; TYPE = type of diabetes; ICC = Intraclass Correlation Coefficient; SEM = Standard Error Measurement. * = DFD related variables; ** = DFU related variables, *k* = Cohen’s kappa coefficient.

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
