# Peer review of "Clinician Assessment Tools for Patients with Diabetic Foot Disease: A Systematic Review"

_jcm, 2020, doi:10.3390/jcm9051487_

Round 1
Reviewer 1 Report
Comments to authors
Many thanks for asking me to review this systematic review, which investigates clinician reported outcomes (ClinROs) and their psychometric properties in the context diabetic foot disease. Whilst there have been a number of reviews which have investigated assessment tools and classification scales in this context (https://doi.org/10.1002/dmrr.3272 ; https://doi.org/10.1111/j.1464-5491.2010.02989.x) this paper arguably provides a more in-depth, novel evaluation of these systems by providing details of psychometric and evaluating the psychometric properties on which they are validated.
Overall the review is well conducted and the authors should be congratulated for their effort collecting the large amounts of data within this study.
Major issues
- Whilst the authors set out to investigate ClinROs in the context of diabetic foot disease, the vast majority of the includes scales are classification systems or risk stratification tools and not outcomes measures. Examples like SINBAD are standardised DFU assessment tools which help to predict ulcer healing. In a research study, tools like this are used in the initial assessment of patients, but are not outcomes measures (Jeffcoate et al make recommendation for reporting standards in DFU studies https://doi.org/10.1016/S2213-8587(16)30012-2). The citation used to define ClinROs (1016/j.jval.2016.11.005) also defines these as tools to define study endpoints. Interestingly the PROSPERO protocol for the study does not mention ClinROs but rather ‘assessment tools’, which is more accurate. Using the term ClinRO therefore seems to be incorrect and makes the review quite confusing. For clarity I strongly recommend the authors revert back to describing in terms of ‘assessment tools’ or remove many of the included scales which are not outcomes measures.
- Although there is a great detail on the raw data for the psychometric properties of individual assessment scales, there is limited description of the individual scales or interpretation as to why some scales may be more validated than others. Furthermore, the review should go into greater depth as to the pros and cons of using certain psychometric properties (e.g. sensitivity) for judging validity. This discussion would really add to the paper and make the review more relevant to clinicians.
Minor points
(Apologies I have not been supplied with have a copy of the manuscript with line numbers)
Introduction
Page 1 - Please revise the term ‘diabetic foot’. Suggest ‘diabetic foot disease’ or ‘diabetic foot complications’.
Page 1 – Please revise the term ‘infeasible’ to ‘unsuccessful’
Page 1 – Please revise ‘partial or total foot amputation (where the entire lower limb might be involved)’ to ‘minor or major amputation’ and provide an appropriate definitions.
Page 1 – The term ‘diabetic patients’ or ‘diabetics’ is generally avoided in contemporary literature. Please revise to ‘patients with diabetes’ or equivalent.
Page 2 – I would argue that a number of previous studies (as described above) have investigated assessment scales in the context of DFU, although have not performed an in-depth investigation of the psychometric properties. Based upon this please provide a more rigorous justification of this study and what it adds to the current literature.
Methods
Eligibility criteria – Please provide a definition for ‘diabetic foot disorders’. What this just ulceration or where other conditions like Charcot’s included?
Eligibility criteria – Please define ‘valid and reliable data’ as this is unclear.
Study selection – Did each of the 3 individual reviewers reviewed all the articles independently or did they only review a set number? As per the Cochrane handbook, at least one reviewer should have reviewed all the articles to provide consistency.
Data extraction – Please revise ‘to make it easier compression results’ as this does not make sense.
Data extraction – If the authors are going to use the COSMIN guidelines for quality appraisal, please provide greater description of the components of the checklist. Without this Table 3 becomes quite confusing as it’s not clear why only some tools are included, whilst many have measurements of reliability and validity.
Data extraction – ‘Variables regarding DFU’ and ‘variables measurable in DFU absence’ are both quite confusing terms. Please revise and suggest the authors give examples.
Results
I am interested as to why the WIfI score was not included in this study, as it is often used in assessing patients with diabetic foot disease and is validated for identifying 1-year risk of major amputation. Please clarify.
Please provide details of the results for COSMIN rating for each scale, as this is not specified (e.g. a table providing details on the ratings).
The authors have identified a large number of assessment scale and I doubt the average reader will have any knowledge of the majority of these. It would be nice to have some more description and detail on the dimensions that each tool studies as this may explain why some have between psychometric properties than other. Suggest each scale is briefly described in a supplemental table. This also adds greater clarity to the ‘Discussion section’, where many individuals scales are explored.
Please clarify if all the studies only investigated patients with diabetes.
Figure 1 – As per the PRISMA flow diagram, please provide the broad reasons for why full-text articles were not compliant with the inclusion criteria.
Tables – Overall the tables are quite difficult to read and interpret. Also having 6 solid pages of tabulated data is not particularly interesting for the reader. I suggest the following revisions:
- Please check the formatting of the tables as some numbers are spread over two rows and are difficult to read (e.g. LR+ for the NIS-LL scale).
- Likewise, whilst some scale names are given with abbreviation, some are only given as abbreviation (e.g. ADA system). Please revise.
- Detailing the type of diabetes seems to be inconsequential but it would be nice to know the sex and age demographics of the population. Suggest this is revised.
- Please clarify the study sample size for the QHRFF scale (19 to 43?).
- Please clarify why the LUMT scale is only described in Table 3 and not also in Table 2.
- Consider moving at least one of the tables into supplementary materials to reduce the bulk of tabulated data. Tables 2 and 3 seem the most relevant, as most clinicians are likely to be most interested in the assessment of patients presenting with diabetic foot disease. Suggest the authors move table 1 into a supplementary materials.
- As per earlier point, further clarification needs to be given to explain why only some scales are included in table 3, as this is not explained.
Discussion – In general there is significant duplication of results from the tables which is unnecessary and not easy for the reader to follow. If the results of individual scales are going to be reported in narrative form, suggest this is moved to the ‘Results’ section.
While the authors describe the psychometric properties of individual scales, there is little discussion as to why differences might occur. It would therefore be interesting to elaborate on the discussion of why some scales are demonstrated greater psychometric properties than others.
Also suggest further discussion on the lack of standardisation of validating these scales and benefits/drawbacks with using certain properties, such as sensitivity and specificity to judge validity (building on the recommendations for research section). Please revise.
The clinical recommendations of the authors are interesting. The QHRFF and LUMB scales are only validated in 19 and 22 patients respectively. Can these scales really be deemed valid when only studied in such a small population??
Please discussion the limitations of the study as this is lacking.
Conclusion
No criteria have been described for the strength of recommendation (e.g. GRADE) so suggest the term ‘strong recommendations’ is revised.
Abstract
No specific issues.
Author Response
Many thanks for asking me to review this systematic review, which investigates clinician reported outcomes (ClinROs) and their psychometric properties in the context diabetic foot disease. Whilst there have been a number of reviews which have investigated assessment tools and classification scales in this context (https://doi.org/10.1002/dmrr.3272 ; https://doi.org/10.1111/j.1464-5491.2010.02989.x) this paper arguably provides a more in-depth, novel evaluation of these systems by providing details of psychometric and evaluating the psychometric properties on which they are validated.
Overall the review is well conducted and the authors should be congratulated for their effort collecting the large amounts of data within this study.
- Authors: We thank you for your comment and for the time spent reviewing this document. We have learned and the manuscript has improved thanks to that. Authors have reviewed the main document and answered point by point the itemized list.
Comments:
Major issues
- Whilst the authors set out to investigate ClinROs in the context of diabetic foot disease, the vast majority of the includes scales are classification systems or risk stratification tools and not outcomes measures. Examples like SINBAD are standardised DFU assessment tools which help to predict ulcer healing. In a research study, tools like this are used in the initial assessment of patients, but are not outcomes measures (Jeffcoate et al make recommendation for reporting standards in DFU studies https://doi.org/10.1016/S2213-8587(16)30012-2). The citation used to define ClinROs (1016/j.jval.2016.11.005) also defines these as tools to define study endpoints. Interestingly the PROSPERO protocol for the study does not mention ClinROs but rather ‘assessment tools’, which is more accurate. Using the term ClinRO therefore seems to be incorrect and makes the review quite confusing. For clarity I strongly recommend the authors revert back to describing in terms of ‘assessment tools’ or remove many of the included scales which are not outcomes measures.
- Authors: Thank you very much for your correction. The authors fully agree that the term "assessment tools" effectively represents much better the concept of instrument on which we have developed in the review, we have replaced ‘clinician-reported outcomes’ or ClinRO by ‘clinician assessment tools’.
- Although there is a great detail on the raw data for the psychometric properties of individual assessment scales, there is limited description of the individual scales or interpretation as to why some scales may be more validated than others. Furthermore, the review should go into greater depth as to the pros and cons of using certain psychometric properties (e.g. sensitivity) for judging validity. This discussion would really add to the paper and make the review more relevant to clinicians.
- Authors: Thank you very much for your comment and suggestion. We have included a table (Annex I) with other details to describe each scale. We fully agree on the fact that in order to make the document more relevant for clinical professionals, it is important to carry out an in-depth study of the psychometric characteristics in order to favour a correct choice of the most appropriate assessment tool. In the final part of the document, in the “Clinical recommendations” section, we have introduced an in-depth study of the main psychometric characteristics included in the study.
Minor points
Introduction
- Page 1 - Please revise the term ‘diabetic foot’. Suggest ‘diabetic foot disease’ or ‘diabetic foot complications’.
- Authors: Thank you very much for your suggestion. We have replaced ‘diabetic foot’ by ‘diabetic foot disease’.
- Page 1 – Please revise the term ‘infeasible’ to ‘unsuccessful’
- Authors: Thank you very much for your correction. We have replaced ‘infeasible’ by ‘unsuccessful’.
- Page 1 – Please revise ‘partial or total foot amputation (where the entire lower limb might be involved)’ to ‘minor or major amputation’ and provide an appropriate definitions.
- Authors: Thank you very much for your correction. We have replaced it as you suggest.
- Page 1 – The term ‘diabetic patients’ or ‘diabetics’ is generally avoided in contemporary literature. Please revise to ‘patients with diabetes’ or equivalent.
- Authors: Thank you very much for your correction. We have replaced ‘diabetic patients’ or ‘diabetics’ by ‘patients with diabetes’.
- Page 2 – I would argue that a number of previous studies (as described above) have investigated assessment scales in the context of DFU, although have not performed an in-depth investigation of the psychometric properties. Based upon this please provide a more rigorous justification of this study and what it adds to the current literature.
- Authors: Thank you very much for your observation. We have considered it in the justification of the study.
Methods
- Eligibility criteria – Please provide a definition for ‘diabetic foot disorders’. What this just ulceration or where other conditions like Charcot’s included?
- Authors: Thank you very much for your comment. We have introduced this definition and its reference.
- Eligibility criteria – Please define ‘valid and reliable data’ as this is unclear.
- Authors: Thank you very much for your suggestion. We have rewritten it to ensure to the meaning.
- Study selection – Did each of the 3 individual reviewers reviewed all the articles independently or did they only review a set number? As per the Cochrane handbook, at least one reviewer should have reviewed all the articles to provide consistency.
- Authors: Thank you very much for the observation. We have rewritten it to ensure to the meaning.
- Data extraction – Please revise ‘to make it easier compression results’ as this does not make sense.
- Authors: Thank you very much for your correction. We have revised this sentence. The manuscript was revised by a proof-reading service (English is not our native language) but surely they ignore this.
- Data extraction – If the authors are going to use the COSMIN guidelines for quality appraisal, please provide greater description of the components of the checklist. Without this Table 3 becomes quite confusing as it’s not clear why only some tools are included, whilst many have measurements of reliability and validity.
- Authors: Thank you very much for your suggestion. We have provided greater explanation of the COSMIN taxonomy in “Eligibility criteria” subsection where is first mentioned.
- Data extraction – ‘Variables regarding DFU’ and ‘variables measurable in DFU absence’ are both quite confusing terms. Please revise and suggest the authors give examples.
- Authors: Thank you very much for your comment. We have replaced both terms by “DFD related variables” and “DFU related variables”.
Results
- I am interested as to why the WIfI score was not included in this study, as it is often used in assessing patients with diabetic foot disease and is validated for identifying 1-year risk of major amputation. Please clarify.
- Authors: Thank you very much for the observation. We agree about the use of WIFI score but we did not find any documents with psychometric or measurement properties. The original study did not include psychometric analysis (Mills JL Sr, 2014. doi: 10.1016/j.jvs.2013.08.003. Epub 2013 Oct 12). Single one validation study in nondiabetic patients was returned in the initial search (Beropoulis E, 2016. doi: 10.1016/j.jvs.2016.01.040. Epub 2016 Mar 16.). With this information, we considered not enough to include it in the results table. If this answer wasn’t well addressed, please, do not to hesitate to indicate us the validation study or any other information.
- Please provide details of the results for COSMIN rating for each scale, as this is not specified (e.g. a table providing details on the ratings).
- Authors: Thank you very much for your suggestion. We did not rate according COSMIN checklist because only 5 scales follow the COSMIN guidelines to address validity and reliability. We have changed the specific sentence to avoid misunderstandings (paragraph 2, Data extraction and results synthesis subsection). Apologies for this confusion.
- The authors have identified a large number of assessment scale and I doubt the average reader will have any knowledge of the majority of these. It would be nice to have some more description and detail on the dimensions that each tool studies as this may explain why some have between psychometric properties than other. Suggest each scale is briefly described in a supplemental table. This also adds greater clarity to the ‘Discussion section’, where many individuals scales are explored.
- Please clarify if all the studies only investigated patients with diabetes.
- Authors: Thank you very much for your correction. We have included a table (Annex I) with other details of each scale (study subjects, dimensions, score, description).
- Figure 1 – As per the PRISMA flow diagram, please provide the broad reasons for why full-text articles were not compliant with the inclusion criteria.
- Authors: Thank you very much for your comment. We have introduced the reasons in the text (first paragraph, results section).
- Tables – Overall the tables are quite difficult to read and interpret. Also having 6 solid pages of tabulated data is not particularly interesting for the reader. I suggest the following revisions:
Please check the formatting of the tables as some numbers are spread over two rows and are difficult to read (e.g. LR+ for the NIS-LL scale).
- Authors: Thank you very much for your correction. We have checked the formatting of the tables.
- Likewise, whilst some scale names are given with abbreviation, some are only given as abbreviation (e.g. ADA system). Please revise.
- Authors: Thank you very much for your correction. We have checked the full names and abbreviation of the scales.
- Detailing the type of diabetes seems to be inconsequential but it would be nice to know the sex and age demographics of the population. Suggest this is revised.
- Authors: Thank you very much for your suggestion. We have added sex and ages dates in the table 4 (Annex I).
- Please clarify the study sample size for the QHRFF scale (19 to 43?).
- Authors: Thank you very much for your correction. We have clarified in the table the three different cohorts of patients included in this study.
- Please clarify why the LUMT scale is only described in Table 3 and not also in Table 2.
- Authors: Thank you very much for your comment. LUMT is a DFU related variables scale which does not show properties like sensitivity or specificity but follows the COSMIN guidelines to address validity and reliability. We have added detail information to clarify it (paragraph 7 in results section).
- Consider moving at least one of the tables into supplementary materials to reduce the bulk of tabulated data. Tables 2 and 3 seem the most relevant, as most clinicians are likely to be most interested in the assessment of patients presenting with diabetic foot disease. Suggest the authors move table 1 into a supplementary materials.
- Authors: Thank you very much for your comment. The three tables include evaluation scales, specifically table 1 are scales address to DFD, table 2 are specific scales for DFU and table 3 includes both types of scales but they are those that have followed the COSMIN methodology in their validation study and, therefore, the psychometric properties are different from the rest.
- As per earlier point, further clarification needs to be given to explain why only some scales are included in table 3, as this is not explained.
- Authors: Thank you very much for your observation. As it was indicated in the previous two points, table 3 includes both types of scales, DFD and DFU related variables scales, but they are those that have followed the COSMIN methodology in their validation study and, therefore, the psychometric properties are different from the rest.
Discussion
- In general there is significant duplication of results from the tables which is unnecessary and not easy for the reader to follow. If the results of individual scales are going to be reported in narrative form, suggest this is moved to the ‘Results’ section.
- Authors: Thank you very much for your suggestion. We have erased the entire discussion section that presented the data in the tables instead of placing it in the results section to avoid duplicating information.
- While the authors describe the psychometric properties of individual scales, there is little discussion as to why differences might occur. It would therefore be interesting to elaborate on the discussion of why some scales are demonstrated greater psychometric properties than others.
Also suggest further discussion on the lack of standardisation of validating these scales and benefits/drawbacks with using certain properties, such as sensitivity and specificity to judge validity (building on the recommendations for research section). Please revise.
- Authors: Thank you very much for your comment. We have introduced just at the beginning of the discussion those both aspects that could condition the difference in the psychometric characteristics presented in this document.
- The clinical recommendations of the authors are interesting. The QHRFF and LUMB scales are only validated in 19 and 22 patients respectively. Can these scales really be deemed valid when only studied in such a small population??
- Authors: Thank you very much for your comment. Effectively, the results based on such a small sample do not have the necessary strength. We have pointed out this aspect immediately after the sentence where these two tools were proposed.
- Please discussion the limitations of the study as this is lacking.
- Authors: Thank you very much for your suggestion. We have added just before the conclusion a paragraph intended to identify limitations that must be taken into account when interpreting the results. If the reviewer identifies another limitation, we will have no problem incorporating it. Thank you very much.
Conclusion
- No criteria have been described for the strength of recommendation (e.g. GRADE) so suggest the term ‘strong recommendations’ is revised.
- Authors: Thank you very much for your comment. We agree with the reviewer that he has made a previous scale and the recommendation was unfounded. We have rewritten the beginning of the conclusion.
Reviewer 2 Report
Clinician-reported outcomes (ClinROs) scales in diabetic foot patients: a systematic review
The main aim of this paper is to investigate clinician reported outcomes (ClinROs) and their psychometric properties.
The authors have explored this in a detailed study.
Although there have been previous assessments of classifications, this present study is novel and adds to previously published material .
Thus it is a comprehensive and useful review.
The paper is well written and the text is clear and easy to read
Although the authors have specifically written a section on “Data extraction and results synthesis”
the statistics in the inter-rater and intra-rater columns are not explained. This section needs to be modified.
CC=Intraclass Correlation Coefficient is not denoted in the legends to Table 1 or 2
What is k in Table 1?
The conclusions are consistent with the evidence and arguments presented. The authors have given useful information on the psychometric properties of clinician reported outcomes (ClinROs).
Author Response
REVIEWER 2
- The main aim of this paper is to investigate clinician reported outcomes (ClinROs) and their psychometric properties.
The authors have explored this in a detailed study.
Although there have been previous assessments of classifications, this present study is novel and adds to previously published material .
Thus it is a comprehensive and useful review.
The paper is well written and the text is clear and easy to read
- Authors: We thank you for your comment and for the time spent reviewing this document.
- Although the authors have specifically written a section on “Data extraction and results synthesis” the statistics in the inter-rater and intra-rater columns are not explained. This section needs to be modified.
CC=Intraclass Correlation Coefficient is not denoted in the legends to Table 1 or 2
What is k in Table 1?
- Authors: Thank you very much for your correction. We have modified “Data extraction and results synthesis” section, included ICC legend and “k” abrevaition.
- The conclusions are consistent with the evidence and arguments presented. The authors have given useful information on the psychometric properties of clinician reported outcomes (ClinROs).
- Authors: We thank you for your comment.
Reviewer 3 Report
I am grateful for the possibility to revise this research study.
Clinician-reported outcomes scales in diabetic foot is a trend topic in the current research literature and may be a main focus of interest for readers.
The title is appropriate.
The abstract sections reflect adequate the main objective of the systematic review.
Introduction may be improved adding new information in order to provide an adequate state-of-the-art including some references. I suggest to include this references include in the attached to complete this requirement because there are more PROMS related to Diabetic foot complications that authors do not included
Chicharro-Luna E, Pomares-Gómez FJ, Ortega-Ávila AB, Marchena-Rodríguez A, Blanquer-Gregori JFJ, Navarro-Flores E. Predictive model to identify the risk of losing protective sensibility of the foot in patients with diabetes mellitus. Int Wound J. 2020 Feb 1;17(1):220–7.
Navarro-Flores E, Cauli O. Quality of life in individuals with diabetic foot syndrome. Endocrine, Metab Immune Disord - Drug Targets. 2020 Jan 28;20.
Methods are well-designed with relevant and complete information. Correct search strategies, good description of the process however there not detailed statistical analyses were included. In order to analyse the level evidence of the results obtained in the research
I suggest authors may improve this aspect for example, should include differences by sex and if it is possible significate differences between different clinometric tools
Tables, figures and redaction of the results are presented in a correct way providing a good presentation of the main finding of the study.
Discussion section include future research studies secondary to the current findings of this study. Clinical considerations, limitations and overall discussion are well-presented
Author Response
- The title is appropriate.
The abstract sections reflect adequate the main objective of the systematic review.
- Authors: We thank you for your comment and for the time spent reviewing this document.
- Introduction may be improved adding new information in order to provide an adequate state-of-the-art including some references. I suggest to include this references include in the attached to complete this requirement because there are more PROMS related to Diabetic foot complications that authors do not included
Navarro-Flores E, Pérez-Ros P, FM M-A, Julían-Rochina I, Cauli O. Neuro-psychiatric alterations in patients with diabetic foot syndrome. CNS Neurol Disord - Drug Targets. 2019 Oct 2;18.
Chicharro-Luna E, Pomares-Gómez FJ, Ortega-Ávila AB, Marchena-Rodríguez A, Blanquer-Gregori JFJ, Navarro-Flores E. Predictive model to identify the risk of losing protective sensibility of the foot in patients with diabetes mellitus. Int Wound J. 2020 Feb 1;17(1):220–7.
- Authors: Thank you very much for your suggestion. We have introduced the references that match with our issue.
- Methods are well-designed with relevant and complete information. Correct search strategies, good description of the process however there not detailed statistical analyses were included. In order to analyse the level evidence of the results obtained in the research
I suggest authors may improve this aspect for example, should include differences by sex and if it is possible significate differences between different clinometric tools
- Authors: We thank you for your comment and we agree with you but with the results obtained, it was not possible to carry out a statistical analysis.
- Tables, figures and redaction of the results are presented in a correct way providing a good presentation of the main finding of the study.
Discussion section include future research studies secondary to the current findings of this study. Clinical considerations, limitations and overall discussion are well-presented
- Authors: We thank you for your comment and for the time spent reviewing this document.
Round 2
Reviewer 1 Report
Thank you for your thorough and thoughtful responses to my queries.
I would recommend a thorough read through to correct English language errors (especially in the discussion section), but otherwise am happy that no further major changes are required.
Author Response
Thank you for your thorough and thoughtful responses to my queries.
Authors: We thank you again for the time spent reviewing this document.
I would recommend a thorough read through to correct English language errors (especially in the discussion section), but otherwise am happy that no further major changes are required.
Authors: After the changes carried out, the manuscript was revised by a proof-reading service a second time (the new certificate was uploaded).